# Reflection on Experiences of First-Year Engineering Students with Blended Flipped Classroom Online Learning during the COVID-19 Pandemic: A Case Study of the Mathematics Course in the Extended Curriculum Program

**Moses Basitere \*, Ekaterina Rzyankina** and **Pierre Le Roux**

Academic Support Program for Engineering in Cape Town, Centre for Higher Education Development, University of Cape Town, Rondebosch, Cape Town 7700, South Africa
* Correspondence: moses.basitere@uct.ac.za

**Abstract:** The shift to remote learning during the COVID-19 pandemic presented a unique challenge for higher education in developing countries such as South Africa, where resources are distributed unevenly. The Department of Chemical Engineering at a South African university of technology used a flipped classroom blended online learning approach in an engineering mathematics course. This study describes the transition of first-year engineering students from face-to-face learning to flipped classroom blended online learning. The Laurillard conversational framework for teaching learning was used to develop the five components of blended learning pedagogy, this allows students to discover, learn, practice, collaborate, and assess (DLPCA). The chosen assessment strategies made use of adaptive learning technology via the WebAssign platform to provide formative and summative assessments as well as timely feedback to each student. The authors examined the remote teaching and learning approach using three indicators: (i) learning—students' learning experiences; (ii) assessment—students' academic performance and integrity; and (iii) students' challenges. The findings had a positive impact on students' learning and performance in the mathematics course. The limitation to the study was that the data were collected only from one university of technology and were not compared with other universities in the country. The overall findings of this study indicate that students require time to adjust to online pedagogy to ensure a smooth transition.

**Keywords:** blended learning; remote learning; engineering education research (EER); WebAssign

## 1. Introduction

The COVID-19 pandemic posed a challenge for higher education in developing nations such as South Africa, where access to resources varies [1]. The transition from face-to-face lectures to online multimodal remote learning sparked concern among students and instructors, as the majority of students at universities of technology come from marginalized communities and previously and currently disadvantaged backgrounds with limited resources, or even limited access to electricity. This problem is not unique to South Africa; other developing nations, such as the Philippines and India, have reported similar problems, such as unstable internet and a lack of electronic devices among a significant portion of their population [2,3]. The transition can be characterized as traumatic [4] for first-year college students adjusting to university life while coping with the COVID-19 pandemic. This has meant that each student required an electronic device to participate in online learning, but the majority of universities lacked the necessary resources. The lockdown restrictions imposed by the South African government as a response to the COVID-19 pandemic impacted all facets of daily life. Still, higher education in South Africa was most impacted, as the traditional method of delivering knowledge to students was no longer viable. Due to the steady increase of COVID-19 cases, of risks, and of local transmission concerns

among students, the first semester face-to-face lectures at the universities of technology were suspended indefinitely. This led to the recommendation that all university faculty courses adopt a multimodal approach to teaching remotely, using either synchronous or asynchronous modes of instruction. Universities in South Africa had limited experience with online teaching and learning, as well as blended learning. universities of technology in South Africa were typically ill-equipped to support extensive online education [5]. The transition to online teaching and learning has been hasty, affording educators and students little time for preparation and few resources. Despite the availability of learning materials and free applications [4,6,7], students have struggled to transition to online teaching and learning, as noted by numerous researchers [4,6]. In addition, lecturers and students experienced anxiety and stress due to an abundance of information and new technology, such as new software platforms, mobile applications, and email. Consequently, academic staff and students must quickly adapt to the "new normal" in the multimodal teaching approach utilized during remote learning. The university's transition to multimodal remote learning was a contingency plan to ensure continuity in teaching and learning. While the shift to multimodal remote learning may have been a good foresight, the digital divide among learners in developing nations with unequal access to resources and digital technologies, such as South Africa, poses a significant challenge [1,8,9].

Several studies have reported on students' acceptance of technology-mediated teaching during the COVID-19 pandemic, and researchers such as Biglan [10] and Neumann [11] have recommended that digital literacy and competency should be taken into consideration when developing a digital learning environment and approaches. Researchers [7,12] have reported that the successful introduction of technology in cases during the COVID-19 pandemic could only be guaranteed if both teachers and students develop appropriate attitudes, beliefs, behaviors, and habits. Scholars [11,13] have reported on favourable E-learning acceptance by teachers in developing countries such as Pakistan during the COVID-19 pandemic lockdown, however weak infrastructure posed challenges with the implementation of the E-learning systems [5].

Due to the unpredictability caused by COVID-19 pandemic outbreaks and lockdown restrictions, universities in South Africa adopted a multimodal approach consisting of three pedagogical approaches: [1] synchronous instruction, [2] asynchronous instruction, and [3] a flipped classroom blended online learning strategy. During the designated lecture slots, the synchronous teaching strategy provides real-time instruction online through Blackboard Collaborate, Zoom, or Microsoft Teams. In this mode, students can ask questions verbally or via chat, and lecturers or course instructors can respond, resulting in an ongoing dialogue between students and lecturers. The asynchronous teaching strategy consists of pre-recording short lecture videos using Zoom or Microsoft Teams and embedding them in a learning management system (LMS) such as Blackboard for students to access at any time. The flipped classroom approach to blended online learning combines synchronous and asynchronous learning strategies, with traditional face-to-face lectures and homework being replaced by pre-class activities such as short, pre-recorded lectures and reading quizzes. Students complete pre-reading and pre-activities such as quizzes, view short pre-recorded videos and participate in collaboration and engagement through WhatsApp social media network sites (SMNS) and use of Zoom breakaway workshops, where students are placed in small groups to create a dialogue in solving problems on a daily tutorial with the assistance of instructor and tutors that are present to respond to students' questions. The classroom time is utilized to reinforce the topics through problem-based discussions and interactive activities. In this study, a flipped classroom blended online approach was utilized and the findings regarding students' experiences with this mode of learning in a mathematics course are presented utilizing Laurillard's model along with a conversational framework for teaching and learning to develop the relevant discover, learn, practice, collaborate, and assess (DLPCA) skills, as described by Lapitan and colleagues [2].

Given the significance of the first year of study and the disruption that the COVID-19 pandemic poses to the first year of study, several questions arise. How did first-year

students perceive the flipped learning strategy? How is transition understood and experienced during the first academic year? What difficulties did first-year students face during COVID-19? This article seeks to address the aforementioned questions through critical reflection and analysis.

*1.1. Conceptual Framework: Laurillard's Conversational Framework*

The study was underpinned by Laurillard's [14] conversational framework for teaching learning. Learning as a conversation distinguishes itself from other theories of experimental and reflective learning and learning through mutual discussion in that it is an implementable model to support learning mediated by technology [15]. The conversational framework is more effective in promoting dialogue between the instructor and the learner in two distinct levels known as the 'discursive level' and 'experiential level' as described by Laurillard [14]. The discursive level, in the traditional sense of the lecturer format, is focused on the instructor and their introduction of conceptual knowledge, ideas, principles, and theory to be learned. This is then followed by dialogue in which the students engage with the instructor by asking questions to clarify any misconceptions. In the mode of the flipped classroom, the discursive level is slightly different to the traditional lecturer format as knowledge, ideas and theory are introduced through pre-recorded videos, which allows the learners to gain the theory and then engage online using Zoom breakaway groups to engage with their peers and with the instructor, who takes the role of a facilitator clarifying misconceptions through dialogue. In this study, the flipped classroom online, blended learning was the most practical teaching approach to adopt during the COVID-19 pandemic as it accommodated both synchronous and asynchronous teaching strategies.

The synchronous approach through Zoom breakaway groups uses the whiteboard option for student engagement and collaborative action to solve mathematical problems. Furthermore, students used the WhatsApp social media platform as an informal learning platform outside the classroom to continue with further dialogue. Both the Zoom and WhatsApp social media options improved students' participation by providing them with space to interact with the lecturer and tutors and to further collaborate to solve problems, unlike in the asynchronous mode where they sat down to watch the video by themselves with no engagement at all. The flipped classroom pedagogical strategy promotes the transfer of new information and enhances students' ability to assimilate and make sense of information. The experiential level encourages interactive activities, allowing students to apply the theory they learned at the discursive level through tutorials, homework, laboratory experiments, and field trips. The instructors continuously observe the interactive activities to monitor students' progress and provide timely feedback that enables students to reflect on the concept in light of their experience with interactive activities and modify their actions as a means of integrating theory and practice. In this study, adaptive learning technology, such as WebAssign, was used for assessment to provide interactive activities. The benefit of WebAssign is that it provides timely assessment feedback, allowing students to reflect on their actions and adjust their learning accordingly.

Sharples [15] used a conversational framework to design the FutureLearn MOOC platform and compare performance metrics across three MOOC platforms. Their results show that adopting a conversational framework promoted higher levels of social engagement with comparable completion rates for the FutureLearn platform. Sharples's [15] work is in alignment with the design in this study, which aimed for an increased level of social engagement mediated by technology such as Zoom breakaways. Furthermore, Neo et al. [16] used a conversational framework to investigate the interaction and communication processes between the students and the teacher-mediated multimedia resources. Their results show that students experienced deep and meaningful learning when communicating and collaborating with each other. Furthermore, their results show that the teacher played a central role in the learning process. Heinze [13] used conversational theory to underpin blended learning and found a theoretical alignment with blended learning. However, their

findings show that there was a need to amend and enrich the conversational framework in order to make it more applicable.

The conversational framework develops and promotes the five components of blended learning described by Lapitan and colleagues [2]: discover, learn, practice, collaborate, and assess (DLPCA). The DLPCA pedagogical strategy was used to design the course using a multimodal approach during the COVID-19 pandemic. The authors discovered that the DLPCA strategy aligns with Laurillard's conversational framework's six modes of learning [14]. The types of learning described by the Laurillard framework are knowledge acquisition, inquiry, collaboration, discussion and practice, and knowledge production. The DLPCA promotes Laurillard's six ways of learning within the conversational framework, except the production of knowledge, which appears to be absent from students' learning. The production of knowledge aligns with the final level of knowledge synthesis in Bloom's taxonomy [8,9]. The authors believe that the level of synthesizing knowledge is an essential component of the engineering curriculum in which students applied the acquired knowledge as part of a report, video blog, and project, for example.

### 1.2. Discover, Learn, Practice, Collaborate and Assess (DLPCA)

The objective of the DLPCA approach strategy was to integrate lecturers (instructors), students, and available technology to address the current problem facing South African higher education during the COVID-19 pandemic. The strategy utilized by the chemical engineering department was an asynchronous, flipped classroom, online blended learning approach. Figure 1 depicts the DLPCA strategy utilized by the chemical engineering department in an online flipped asynchronous, blended, and synchronous mathematics course.

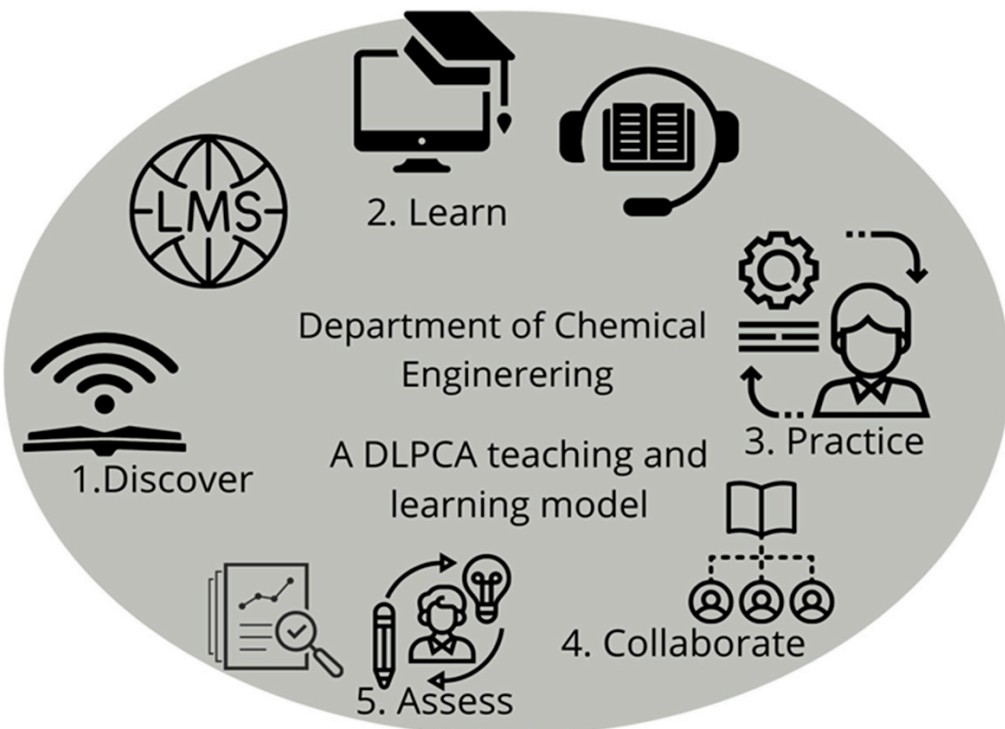

**Figure 1.** Flipped teaching and learning model.

In addition, Table 1 details the presence of technology and type of learning in DLPCA.

**Table 1.** Examples of digital tools and learning types of DLPCA.

| Learning Type | Examples of Related Digital Tools |
| --- | --- |
| Discover | The students were asked to discover resource materials containing lecture videos, tutorials, lecture notes assigned for a specific week, and WebAssign registration for formative (quiz) and summative (exam) assessments on the Blackboard learning management platform. |
| Learn | Students were expected to read the notes first, watch short, pre-recorded videos, and do a daily tutorial exercise. |
| Practice | The practice component with the assistance of Zoom breakaway groups using a whiteboard and WhatsApp social media. |
| Collaborate | Students use Zoom breakaway group sessions and WhatsApp social media to collaborate, and they ask questions and respond to each other with the assistance of tutors and course instructors. |
| Assess | The assessment for both formative and summative assessment was achieved using WebAssign. |

## 2. Materials and Methods

In this study, researchers examined the remote engagement and learning of chemical engineering students in a mathematics course using a flipped classroom, online blended learning pedagogical approach. We used a questionnaire for this study because it offers the advantages of standardized and open responses to a variety of topics. The online survey is dependable, valid, quick, and simple to complete [17]. For this questionnaire, the researchers used several types of question, including dichotomous questions, multiple-choice questions, and open-ended questions. It is crucial that question types are appropriate to their purpose, i.e., sufficiently focused and concrete (rather than, for instance, too general and abstract), yielding actionable and relevant data [17] (p. 475). We designed the questionnaire for this study with these factors in mind.

### 2.1. Participants

The participants in this study were first-year students enrolled in a mathematics course in the department of chemical engineering at a university of technology (UoT) in South Africa. These students utilized the Calculus (8th edition) e-textbook by James Stewart, published by Cengage. Students were automatically enrolled in the official UoT Blackboard learning management system (LMS) upon registering for a mathematics course. Students were also required to create Cengage student accounts to access the WebAssign platform. Through the free online platform WebAssign, students gained hands-on experience in learning to master engineering concepts. The publisher provided access to the e-textbook and to WebAssign for a six-month testing period in 2020. WebAssign is used for formative and summative assessment in the classroom. Students were sent a link to a WhatsApp group where they could join lecturers, tutors, and mentors who could provide support in their learning.

### 2.2. Ethical Consideration

An informed consent form was embedded into the online questionnaire before the student completed the questionnaire to obtain permission to use the data for research. If a student felt uncomfortable, he or she could withdraw from the survey at any time or decline to answer specific questions. When collecting data for this study, the researchers maintained privacy, confidentiality, anonymity, and non-traceability. The researchers utilized pseudonyms in student transcripts for data analysis and presentation.

*2.3. Research Instrument: Case Studies*

For this study, researchers chose a case study [18] instrument because it provides a unique illustration of real people in real situations, allowing readers to comprehend the process of collecting and analyzing data. Instead of evaluating and testing surface phenomena, a case study requires researchers to collect a small amount of in-depth data pertinent to the purpose of this research. A case study can provide readers with real-world problem situations, and abstract concepts can be incorporated into theories [17]. In addition, the aforementioned case study can present situations in ways that would not be possible using a quantitative paradigm, as the numerical responses can explain a person's feelings and thoughts regarding specific situations or questions. A case study also accepts multiple variables operating within a single research case and is a tool that can be used for mixed-methods research.

In this research case study, a questionnaire was utilized as both a quantitative and qualitative data collection technique. Instead of selecting one of the multiple-choice responses, students could also briefly explain the selected response or elaborate on the difficulties they encountered during online remote learning.

*2.4. Procedure*

Before sending the link to students, the researchers conducted a pilot questionnaire. The primary purpose of the pilot questionnaire was to improve the questionnaire's reliability, validity, and usability. Piloting questionnaires also allows for an opportunity to improve the accessibility and readability of the questionnaire on internet-capable smartphones. The questionnaire was tested before conducting the research project as the majority of first-year students in South Africa come from marginalized communities and have limited use of smartphones as a digital study tool. After piloting the questionnaire, the researchers distributed the link to the online LimeSurvey questionnaire via WhatsApp groups and the official UoT LMS to all the students.

LimeSurvey was chosen for this study because it offered complete confidentiality and anonymity for participants and was not connected to the private emails of the researchers and students. The length of the questionnaire was limited to 20 min to avoid interfering with students' study and personal time. The link invitation was sent to all first-year students in their department of chemical engineering. The questionnaire was available for two months to allow sufficient time for completion. There were a total of 65 students who participated in this study; however, not all responses were usable due to missing values and incomplete responses. The researchers only used the responses of students who completed the questionnaire in its entirety (*n* = 27). Participants who answered fewer than half of the questions in LimeSurvey were excluded from the data analysis phase.

## 3. Results and Discussion

The goal of this study was to assess students' perceptions of flipped classroom, online blended learning in a mathematics course. Furthermore, the advantages and disadvantages of this pedagogical approach as a response to the COVID-19 pandemic were determined. The researchers were guided by various questions and themes based on the DPLCA pedagogical strategy to evaluate student experiences and to improve the course in the future. The researchers discussed the major themes that emerged during the data analysis in this section. To begin, the researcher analyzed the participants' descriptions in terms of how they relate to technology. The researchers elaborated on the main components of the DPLCA framework, as well as academic integrity for conducting online assessments. Finally, the researchers discussed the participants' challenges in transitioning from face-to-face teaching and learning to a fully online environment. An analysis of participants' descriptions in terms of access to the technology was also undertaken.

### 3.1. Participants' Description before They Enter Univerisity in the Context of COVID-19

Many public universities in South Africa have extended curriculum programs (ECP). The primary objective of an ECP is to improve the academic performance of first-year students who, despite meeting minimum admission requirements, are at risk of failing, dropping out, or taking an extended amount of time to complete their academic qualification. The ECP primarily attracts students from lower income groups and who have limited resources, as evidenced by the questionnaire responses of students. Figure 2 depicts the questionnaire question regarding the relationship between the age of students and when they first had access to any type of technology ('Any type of technology' means access to any digital devices (computer, laptop and mobile phones) or software as well access to the internet (Wi-Fi or mobile data) before entering university.

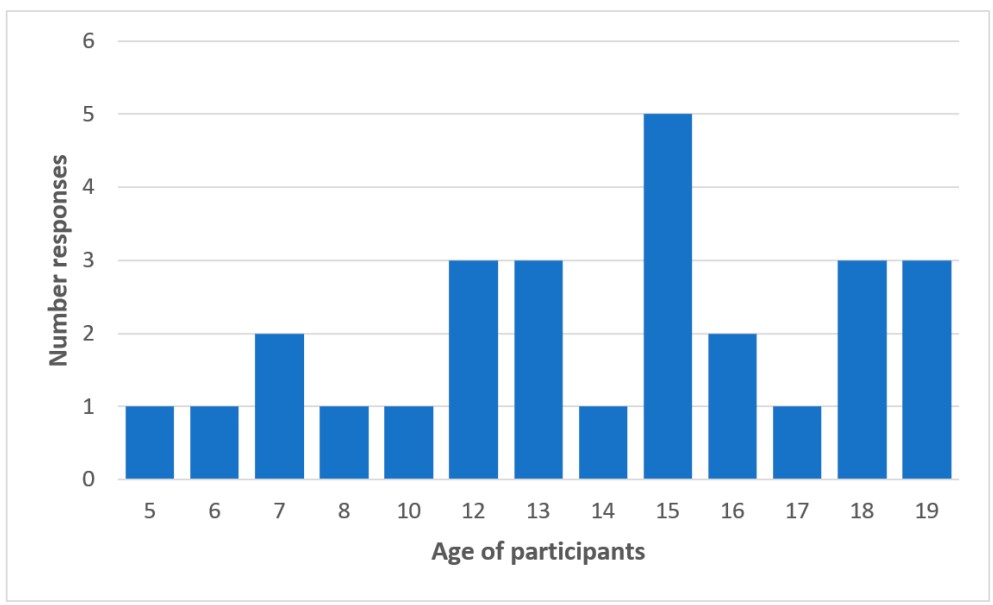

**Figure 2.** Correlation between age and computer access at school before entering university.

The total number of responses received was *n* = 27. At age five, only one of the participants in Figure 2 had access to a computer or any type of technology. Five students reported being introduced to or having access to technology, such as a computer device, at the age of 15 or later (at the ages of 16—two students, 18—three students, and 19—three students), indicating that university was the only place where they had access. Some students reported being exposed to higher levels of computer literacy and advanced computer skills, such as programming, while in school. This was supported by the following quotation:

> Student A: *"I was already learning to use different software and loved to be on the computer but mostly those years to play games. But I did Information Technology after I immediately matriculated in 2015, so I have advanced computer experience". [sic].*

In terms of computer literacy, Student A was better prepared for remote learning at university and will likely have less difficulty adjusting to and overcoming obstacles associated with distance education than other students. Student A stated in his quote that he has advanced computer experience that is not necessarily related to programming. This means that the student has worked with any type of technology and has taken computer-related classes in school. This student's quotation indicates that, by knowing technology, they may be technologically advanced in comparison with other students in the same class, but knowing technology and how to navigate technology and software does not guarantee that this student will be academically advanced.

Figure 3 shows participants' experiences with online or distance learning before they entered their university. The results of the survey question, "Did you experience online

learning before COVID-19?" indicate that the majority of the participants (21 out of 27) had not experienced online learning before the pandemic.

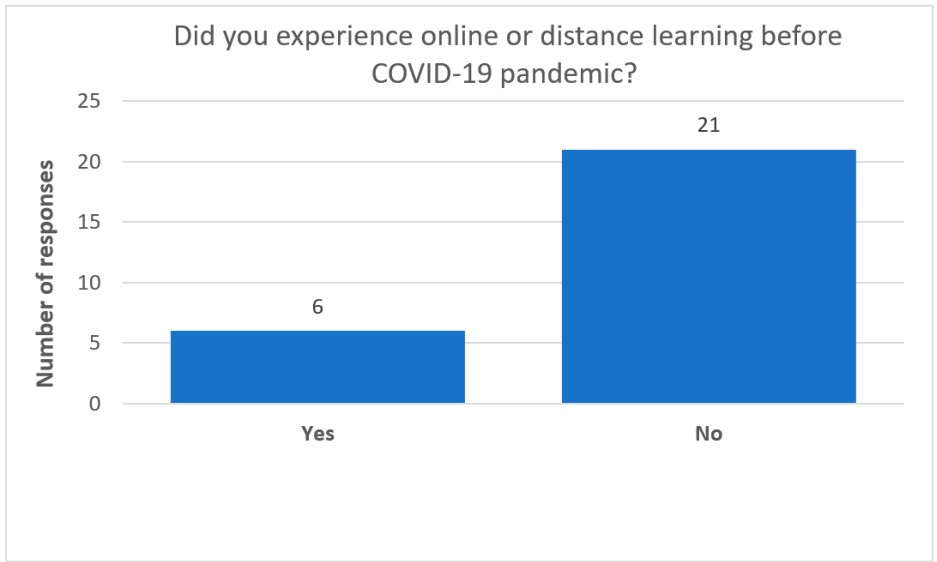

**Figure 3.** Previous experiences with online/remote learning of the participants.

This suggests that the participants' had not been exposed to virtual schooling prior to the outbreak of COVID-19, and that the transition to online learning was a relatively new experience for them. The participants' likely experienced a steep learning curve as they adjusted to this new type of education. The remaining seven participants had experience with online learning prior to the pandemic, which might have helped them adapt to the sudden shift in education more easily.

From these data (see Figure 4), it appears that most of those who responded to the question spend between three and six hours per day for study purposes. Ten respondents reported spending three hours per day, seven reported spending between five and six hours per day, and three reported spending more than six h per day. This indicates that most respondents spend between three and six hours per day for study purposes. However, it is also worth noting that three respondents reported spending less than two h per day for study purposes, indicating that there may be a small portion of respondents who spend less time studying than the majority.

### 3.2. Discover: Access to Knowledge and Learning Material

After each Zoom session, all recorded materials (video and whiteboard slides) were uploaded to LMS Blackboard. Students also received lecture videos and course materials from lecturers. In addition, some students used other platforms (YouTube, etc.) for additional information about the lecture material to help comprehend the course material.

*Student G: "I watch videos on YouTube or any other learning resources". [sic].*

In prior studies [14,19], it has been reported that the use of pre-recorded videos had a positive impact on teaching and learning as part of the flipped classroom blended learning before the pandemic. As described by Laurillard in her conversational framework [14], when students discover learning materials through handouts and pre-recorded videos, they acquire knowledge. The purpose of activities that lead to the acquisition of knowledge is to provide students with essential information or concepts that they must learn and comprehend.

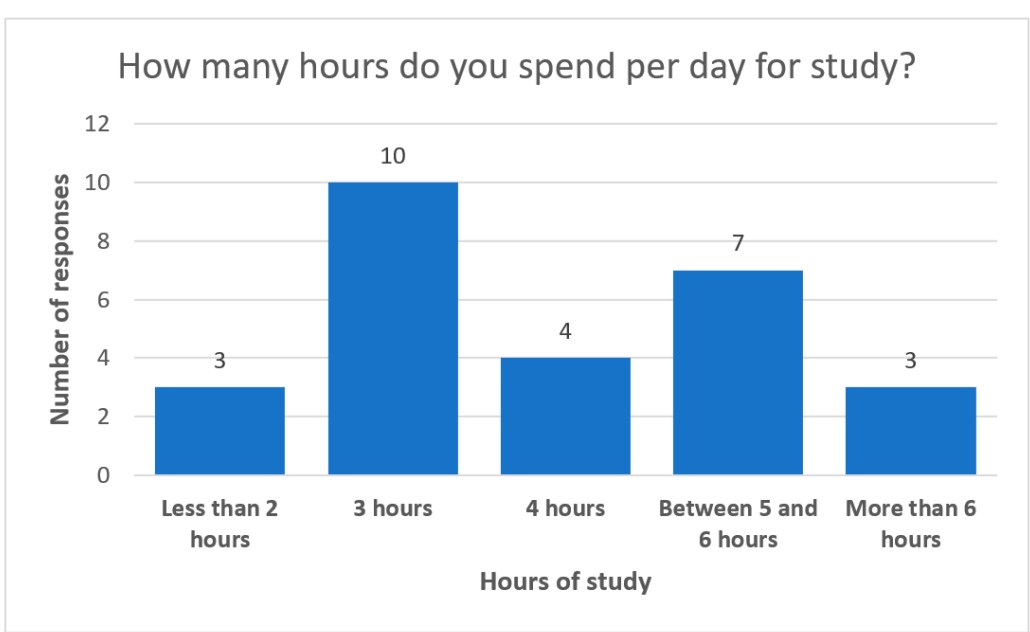

**Figure 4.** Allocated hours per day for study.

### 3.3. Learn: Asynchronous, Synchronous Learning

The department of chemical engineering implemented flipped classroom online blended learning with WebAssign and an interactive calculus e-textbook. The mathematics teaching was undertaken asynchronously using short, pre-recorded videos of chapters uploaded to the learning management system, Blackboard. During classroom exercises and mathematics workshops, students engaged with the instructors via a Zoom breakaway session with an online whiteboard. In addition, students had access to support mechanisms through the social media site WhatsApp, where lecturers, tutors, and mentors interacted with students. The role of tutors was expanded to include assistance with academic matters, and the role of the mentor was to assist students with the mental challenges associated with distance learning.

The students felt that they required additional assistance from the instructors to fully comprehend certain concepts. Despite negative experiences with remote learning in other subjects, students identified some positive aspects of online education:

*Student E: "Certain concepts are difficult to explain to someone immediately. Though with time one can record a video to explain better". [sic].*

The need for student support is related to Vygotsky's zone of proximal development (ZPD) [20], in which learners require the assistance of a more knowledgeable individual to progress from unfamiliar to familiar concepts [20]. In this investigation, students were given mathematics problems in which they were required to apply previously acquired or discovered knowledge to solve engineering problems.

### 3.4. Practice: Mastering the Engineering Concepts

Students were permitted to complete classwork problems via Zoom breakaway groups and WhatsApp. Following the pre-recorded videos, the instructor also assigned textbook-based exercises for students to complete. The pre-recorded videos were also used for exam/test preparation by the students, and free access to the videos allowed for some learning flexibility.

*Student L: "The practice time would differ depending on my understanding of a certain topic. I mean like you can put your 2 h but if it passes and you're still struggling on the topic then you'd have to adjust your time. [sic].*

### 3.5. Collaborate: Learning Collaboration

In the mathematics course, students could pose questions directly to the instructor by activating the microphone or writing in a chat box in the Zoom breakaway room, as well as via the WhatsApp social media platform. In addition, students were pleased with the approach in the mathematics course, where the instructor primarily used WhatsApp to assist students with course-related questions and administrative issues. The interaction and collaboration by students via WhatsApp are highlighted by the following four student–participant comments.

*Student F: "Students prefer to use any online collaborative tools. I had WhatsApp messaging, video or audio calls with classmates and lecturers and mentors". [sic].*

*Student M: "There is less pressure to interact with other students, as you have the help of devices, apps, and multimedia tools to make learning a more interactive and enjoyable prospect." [sic].*

*Student L:" I have communicated on homework with my classmates via Whatsapp messaging." [sic].*

*Student A: "Me and my friends have been helping each other with some of the things we didn't understand". [sic].*

These four students' responses suggest that digital tools and apps enable students to interact with their peers even when they are not physically together. Student M's comment about the lessened pressure to interact with other students implies that digital tools make it easier for students to communicate and collaborate. Student L's comment about using the WhatsApp tool to communicate on homework exercises (as shown in Figure 5) suggests that digital communication tools are being used to facilitate group work and collaboration.

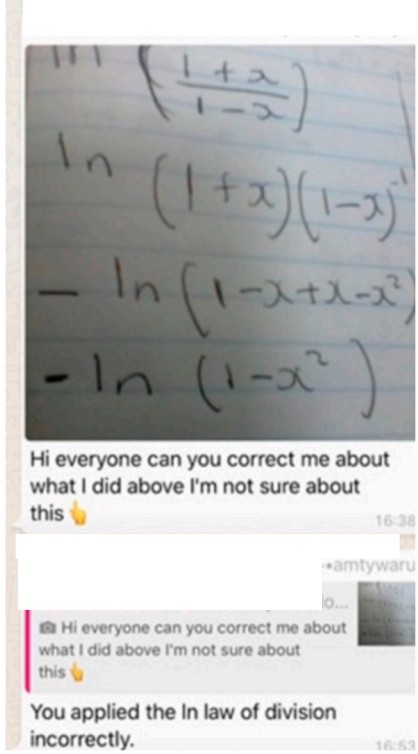 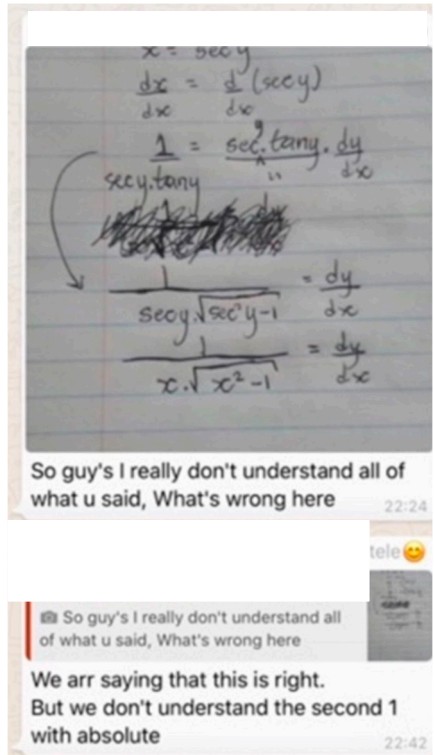

**Figure 5.** Two WhatsApp screenshots of a conversation between chemical engineering students.

Other studies have also reported that the use of the WhatsApp social media increased student engagement and collaboration and improved student learning [21] and performance in mathematics classes [22]. Widiasih [23] has reported that the WhatsApp social

media network supports distance learning through online activities for physics classes. This is consistent with the, concept of community of practice, in which students collaborate towards a common goal to attain a learning outcome [7,19].

Finally, Students A and F's comments about helping each other with things they did not understand further reinforces the idea that digital tools allow students to connect in meaningful ways even when they are not physically together.

### 3.6. Benefits of Online Learning

Figure 6 illustrates participants view/s on the benefits of online/remote learning. To support the findings from this graph, we included participants quotations below.

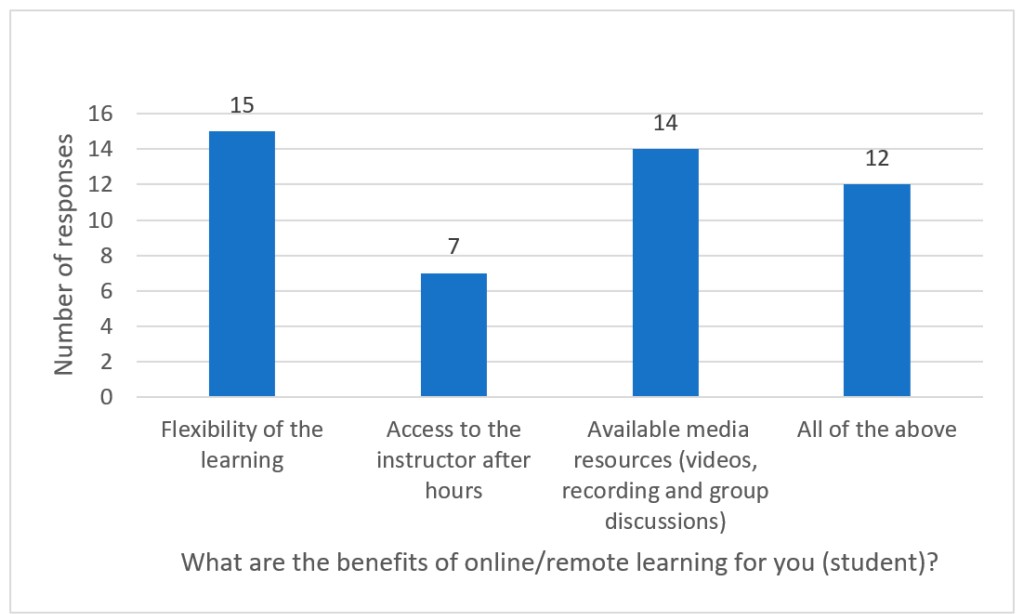

**Figure 6.** Participants' views on the benefits of online/remote learning.

The results of this survey reveal that flexibility of learning is the most popular advantage of remote learning, with 15 responses. This suggests that many students prefer the freedom to learn on their own schedule, rather than having to attend a traditional classroom setting. Access to the instructor after hours was the second-most popular advantage, with seven responses. This indicates that many students appreciate the convenience of having instructors available outside of normal business hours, which may be especially beneficial for those with jobs or other obligations. Available media resources (videos, recordings, and group discussions) were the third-most popular advantage, with 14 responses. This suggests that many students find these resources to be extremely helpful in understanding the material, especially when they are unable to attend a traditional classroom. Finally, 12 of the responses indicate that the students perceive all of the above advantages of remote learning. This suggests that many students believe that all of these advantages are important and beneficial to the remote learning experience.

Researchers were presented with the below students' views regarding the benefits of online learning. For example, Student B: *"We can learn using different sources at the same time, such as books, pc, and cellphones". [sic].*

Student B's statement suggests that they believe in the power of multiple sources of information when it comes to learning. By utilizing multiple sources, such as books, computers, and cellphones, Student B believes that it is possible to gain more knowledge.

Student D: *"You learn to study on your own". [sic].*

Student D's statement points to the importance of self-study. Student D believes that in order to learn effectively, one must take the initiative to learn independently.

> *Student E: "Our lecturers made this online learning more bearable for us, it wasn't as challenging as I was expecting it all because of the support they provide. Those replies to the queries meant a lot". [sic].*

Student E's statement highlights the importance of support from their lecturers regarding online learning. Student E believes that their lecturers have made online learning more bearable by providing support and responding to queries. This shows that support from instructors is essential for successful online learning.

### 3.7. Assess and Academic Integrity for Online Assessments

The issue and debate on academic integrity for remote online learning assessment has become more prominent among researchers, practitioners, and teachers worldwide. During the beginning of the lockdown, lecturers transitioned to online remote learning and experienced a heavy workload in creating lecture videos and multimodal content. In regard to exams and tests, lecturers faced the new question of how to create assessments in a way that was fully protective of academic integrity in the online environment. Instructors encountered difficulties with student access to technology (devices and cameras) and data, while also having to evaluate students' technological proficiency as discussed in Section 3.1.

> *Student B: "As students, we cheat ourselves sometimes. I have seen it and tried it before, I think it was my first physics quiz. I failed the 1st try then asked for online help. After that, I felt guilty in a way because I realized I knew nothing and my time was just wasted so from then on I worked hard to understand the content . . . I saw similar patterns from other students who would ask me questions while we were writing tests during this academic year . . . So overall some marks might not be legit for other students including my only one test quiz. This should be scary enough to say that students might not be deserving to move further especially in this way moving forward". [sic].*

In the face-to-face mode of instruction, lecturers have complete control over their classroom instruction and academic integrity. In online remote learning, academics have less control over the teaching and examination process. The system for online examinations of students requires additional safeguards and monitoring to prevent cheating. As stated previously, South Africa and other third-world countries have faced similar challenges in academic teaching and learning practices. Due to the absence of safeguards in the system, as indicated by Student B's remark, cheating has become even more prevalent. This study used WebAssign to examine students' multifaceted understanding of cheating prevention. Additionally, researchers employed the following examination parameters: password-protected tests and assignments, double randomization (randomization by question order and by student), and establishing times for exams. In this study, researchers did not monitor students working with cameras due to a lack of high-tech equipment and limited data available to the students.

There are additional effects of online education on academic and professional work in higher education, as discussed in Beethamet's [24] paper. During the pandemic and emergency online teaching response, academic teaching has (necessarily) taken on more transparent forms, such as benchmarks for online learning materials and time limits for uploading recorded lectures, as noted by Titus [25].

### 3.8. Challenges of the COVID-19 Transition

In this last section of this study, researchers identified limitations in the responses of students. Chemical engineering students highlighted the lack of uniformity in the provision of remote learning for other subjects. In some courses, there was no interaction with the instructor, and students experienced frustration, as discussed by students C and D below:

> *Student C: "Some teachers don't know how to teach online, and it takes longer than it should because of their teaching strategy." Anytime you want to get what the lecturer was saying, you simply go to the videos". [sic].*

*Student D: "Some concepts need elaborating and we cannot understand everything from the videos we get". [sic].*

These two quotes demonstrate that the transition from face-to-face to remote learning was not simple for either students or for course instructors. The majority of instructors had to begin teaching online for the first time, resulting in a poorly structured course offering that did not meet the needs of all students. In addition to the pressure to adapt to the new method of instruction, some students also encountered academic difficulties.

*Student F: "With remote learning, you need more hours to study because you are by yourself". [sic].*

In contrast to students C and D, Student F felt that, during remote learning, there was less human interaction between students and lecturers, and they needed more time to undertake self-study. Perhaps this student learns better when surrounded by other students, and perhaps Student F needs more time to learn how to study remotely.

*3.9. Integration and Impact of Online Learning Technologies into the Curriculum during the COVID-19 Pandemic*

With the ongoing growth of online learning, curricula will incorporate online learning more frequently. Careful planning and preparation will be necessary to ensure that students are appropriately equipped to utilize online learning tools.

The utilization of online and digital teaching and learning tools has the potential to contribute positively to the sustainability of education. This is because it can lessen the environmental effect of conventional modalities of education, such as commuting and paper use. The utilization of pre-recorded videos and the incorporation of many digital platforms, such as LMS Blackboard, Zoom, and WhatsApp, provides students with a flexible and easy learning environment.

The flipped classroom blended learning strategy developed by this chemical engineering department has the potential to improve overall learning outcomes by allowing students to gain knowledge at their own pace using pre-recorded lectures and handouts. In addition, the strategy gives students possibilities to communicate and interact with their peers and instructors via digital platforms, thereby boosting their learning experience.

Yet, the unfortunate shift from traditional to online education imposed restrictions on many South African student groups. The use of digital technologies, for instance, presents obstacles, notably in providing equal access and equitable learning outcomes for all students, particularly those from underprivileged backgrounds. In addition, the lack of face-to-face interaction might result in feelings of isolation and disengagement among students, as demonstrated by the poor experiences with distant learning in other areas. In order to maintain a sustainable and fair education for all students, it is essential for educators to evaluate the ramifications of incorporating digital technologies into their lessons, including the potential benefits and negatives.

Despite the fact that online teaching and learning has the potential to help for sustainable efforts in higher education, it is crucial to ensure that online learning is equitable for all students, especially those from low-income groups. This demands continuous investment in the development and refinement of online learning tools and support structures, as well as initiatives to bridge the digital divide and ensure fair access to technology and resources.

## 4. Conclusions

The COVID-19 pandemic altered many aspects of education. Transitioning from face-to-face to online learning necessitated that both students and instructors have access to and are proficient with digital technology to improve learning through dialogues, as described by Laurillard's framework. Our findings reveal that not all students had access to technology, with some gaining access only after enrolling at a university of technology. As result, the familiarization of new technology delayed the learners' focus on gaining mathematical knowledge as they spent time familiarizing themselves with the new technology instead.

Academic integrity in online education is also made more difficult by restricted access to high-tech devices and expensive interactive platforms, such as WebAssign. Incorporating the WebAssign platform into the teaching and learning process positively affected students due to the visual modes used to explain engineering concepts.

The findings suggest that lecturers should carefully plan the transition from face-to-face to a fully online mode because not all students are prepared and/or have prior experience with technology. This suggests that the foundation or preparation courses be included in the curriculum before students engage with online learning resources. This study should not be considered a method of online/remote learning; however, it may be used as an example of online learning that uses asynchronous and synchronous methods by other researchers and practitioners as they design engineering calculation-based subjects in a developing country.

Furthermore, the findings of this study on flipped classroom blended online instruction for first-year engineering students at a university of technology with limited resources can be utilized to organize learning activities and develop curricula for countries with comparable settings and challenges. In addition, the study contributes to the literature on the effectiveness of adopting a flipped classroom blended online learning method to facilitate the transition of first-year engineering students to blended learning in resource-constrained settings such as South Africa. In addition, the study highlights the significance of utilizing adaptive learning technologies for formative and summative assessments to provide students with timely feedback, which can then contribute to the remarkable course achievement of students. Finally, the study offers insights into the obstacles and opportunities of remote learning in higher education and implies that students can succeed in online blended learning environments if they are provided with the necessary pedagogical approaches and evaluation strategies.

**Author Contributions:** Methodology, E.R.; Software, M.B.; Formal analysis, E.R.; Investigation, M.B. and E.R.; Data curation, M.B., E.R. and P.L.R.; Writing—original draft, M.B. and E.R.; Writing—review & editing, M.B., E.R. and P.L.R. All authors have read and agreed to the published version of the manuscript.

**Funding:** This research received no external funding.

**Informed Consent Statement:** Informed consent was obtained from all subjects involved in the study.

**Conflicts of Interest:** The authors declare no conflict of interest. The funders had no role in the design of the study; in the collection, analyses, or interpretation of data; in the writing of the manuscript, or in the decision to publish the results.

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
