# Peer review of "Reflection on Experiences of First-Year Engineering Students with Blended Flipped Classroom Online Learning during the COVID-19 Pandemic: A Case Study of the Mathematics Course in the Extended Curriculum Program"

_sustainability, doi:10.3390/su15065491_

Round 1

Reviewer 1 Report

This case study focuses on the review of a flipped online teaching course delivered during the pandemic. The course design is based around Laurillard's conversational learning model. The strength of the paper is in the way that it describes how online (asynchronous and synchronous) technologies may be used to support a flipped design approach for remote learners with varying levels of tech access. The paper can be improved though in a number of ways.

First, the literature review could be more detailed in addressing the extensive studies that have been conducted previously on conversational learning and how this addresses aspects of learning community (e.g. Sharples & Ferguson, 2019). The learning community dimension to student engagement in this course experience is not really tackled, but is an important one to explore in the discussion from the perspective of inclusion and student support. The other key area of literature that is missing is around technology acceptance and strategies for student engagement with online learning designs. There is an extensive range of publications of socialisation measures for students to support their engagement in online learning. The observation in the conclusion that the transition phase is not straightforward and has contingent factors is already well established in the literature.

There are some missing elements in the reporting of the data. The total number of surveys that were completed out of the course population needs to be stated. It would be helpful to have some discussion on the technology and learning profiles of the cohort prior to embarking on this course. To what extent did learners with prior technology experience and/or familiarity with flipped learning methods adapt more easily to the target learning methods? Was any statistical analysis of the survey results conducted, based on the profile of the cohort? The survey results are not presented in full and it's hard to judge to what extent the discussion of them is a fair representation of the data. 

It would be interesting to compare the survey data with analytics on the time that student spent watching the videos, to see if there is any correlation between acceptance / satisfaction with the learning methods and time on task. Equally it would be helpful to present some analysis of the WhatsApp chats in terms of the range of participants and nature of the discourse to learn more about how this component of the online learning experience. By addressing these points, a rich picture of the student learning experience could be presented. At present the paper only offers a selective picture of the learning, based on some themes that the authors have identified from the survey data.

The conclusion is disappointing and what's lacking is discussion of the significance of the reported design approach and how this might influence future teaching strategies. What are the transferable lessons learned from this case illustration and how might they apply to other educational contexts? Spell out the significance of your findings to the reader.

With attention to these points the paper could be an interesting contribution to the literature on conversational learning design.

Author Response

Dear Reviewer

Thank you very much for reviewing our paper and we believe it has improved the paper.We have attached the author's to show all the correction made.

Kind regards

Dr Moses Basitere

Reviewer 2 Report

Manuscript sustainability 212193-"Reflection on experiences of first-year Engineering students with blended flipped classroom online learning during the  Covid-19 pandemic: A case study of the mathematics course in  the Extended Curriculum Program."

This study describes the transition of first-year engineering students from face-to-face learning to blended online learning with the flipped classroom. Laurillard's conversational framework for teaching learning was used to develop the five components of blended learning pedagogy that enable students to discover, learn, practice, collaborate, and assess. The assessment strategies chosen made use of adaptive learning technology through the WebAssign platform to provide formative and summative assessments as well as timely feedback to each student. The authors examined the distance teaching and learning approach using three indicators: (i) Learning: students' learning experiences, (ii) Assessment: students' academic performance and integrity, and (iii) Students' challenges. The results had a positive impact on student learning and exceptional performance in the Mathematics course. The overall results of this study concluded that students needed time to adapt to the online pedagogy and make the transition smoothly.

To improve their work, I suggest.

1.     Includes in the abstract section the most relevant results and a paragraph with the limitations of this research and future work.

2.     Check the English grammar and spelling of the entire paper.

3.     Include in the keywords blended flipped classroom.

4.     Apply the appropriate style in table 1 for MDPI journals

5.     Improve the resolution and quality of figure 1.

6.     Summarize the method in an outline or concept map.

7.     To make your research more impactful, place the study data and analysis in a dataset; you can create one at https://data.mendeley.com/.

8.     Improve the resolution and quality of figure 2.

9.     Improve the discussion section by comparing your contributions with other authors.

10.  Reinforce the conclusions section.

11.  Update the references of the 24 references 50% are out of the five years of validity.

12.  Apply the journal's citation and referencing style.

Author Response

(The authors gave the same response as above.)

Reviewer 3 Report

This study is focused on using DLPCA framework in distant learning. The idea is good and important. The topic is relevant in the field, I am not sure, if it is original. The benefit is usage it in specific conditions of developing country with limited access to computers and even electricity. Unfortunately, these specific conditions were not taken into account in the research.

It does not add too much to the subject area compared with other published material, because readers only have the conclusions of authors. It is not possible to see the students´ responses, even the questionnaire was not published.

The methodology is correct, used questionnaire was piloted. But authors present only conclusions, not the data. And the one or two citations of students’ answer cannot be considered as appropriate source or argument. There are not data to assess if conclusions are consistent with the evidence, although arguments are address to the main question.

The work with sources and references has to be improved. In the text the references are not following the logical order.  After references 4, 5 and 6 in 54th line is reference 23 (134th line), then 13 (160th line), references 15 and 16 are for the first time used in lines 353, resp. 356. References from 7 to 12 are not in the text.

Self citations Nr. 9 (is only in References list), 18 and 19 I consider inappropriate.
In the article is written that the sample is 65 students and some of them were excluded because of incomplete questionnaire. The information about the number of used questionnaires misses. But it is not a big problem, because authors present data to only one question. Even so, the presentation has to be improved. In the picture are 2 respondents at age 5, but the text writes about 3 of them (line 243). There is also a mistake in the picture – „Age of responses“.

Line 240 „any type of technology“ It is not define, what the authors means. Technology is not only computer, but also e.g car or pump.

Author Response

(The authors gave the same response as above.)

Round 2

Reviewer 1 Report

Thank you for responding to the feedback for the first review of your paper. The additional contextual information that you have provided has improved the readability of the paper. I would still suggest that some improvements can be made to the draft, which would clarify the contribution that your paper is making to the literature, whilst also enhancing its readability.

The paper lacks transferable lessons learned that can be applied to other contexts. The key conclusion that students need to be supported and prepared well before making the transition to online learning is well established and should be acknowledged as such – please reference the extensive literature on digital learning when making these points. Could more be said here about the challenges of doing this within developing countries, addressing the equity and access to technology issues, as well as the required levels of pedagogical support provision to students? This would help to draw out the key contribution that your study is making to the literature on student transition to conversational online learning.

A reference is made to academic integrity on p/14 (l. 527), but is this what is intended? The context suggests that you are referring to equity and inclusion issues, as opposed to the academic conduct of your students.

In drawing conclusions from the paper, it is advisable to reference the limitations to this study and to the research methods and to suggest how future studies might build on this case study work. This helps to put in context the value of the conclusions that you are sharing. It is not enough to mention a limitation to your research methods in the abstract to your paper, but then not acknowledge this in the discussion and concluding comments.  

The draft needs extensive proof reading. Here is just a selection of items for attention.

Some sentences could be clearer – e.g. in the abstract: The findings indicate that the flipped course design….

Page 1 line 35: the text in inverted commas is not referenced

Page 2: line 68 onwards – name the researchers in the text.

Page 2. lines 82-83: Blackboard Collaborate, Zoom, MS Teams

Page 4: How does your design build on the work of Sharples. The connection needs to be more clearly expressed.

Page 7: the new paragraph includes a variety of tenses, present, past and future tenses and is confusing – this needs to be reviewed.

Page 10: ‘The majority of the need for…’: this does not make sense and needs to be rewritten

Page 12: missing apostrophes  e.g. participants’; students’

Page 12 “Our lectures” (sic)

Page 13 line 494: this sentence does not make sense

With attention to these points the draft will be much stronger.

Author Response

Dear Reviewer

Thank you very much for taking time to review our work.

Kind Regards

Moses

Reviewer 2 Report

Remove figure 5 or improve the quality and resolution, it does not look good.

To strengthen your research you can place the data, tables, analyses and figures in an open dataset for example at https://data.mendeley.com/.

Author Response

Thank you very much for taking time to review our work.

Kind Regards

Moses

Reviewer 3 Report

Also in the 2nd version are problems with references. In the list of references the sources are not listed alphabetically, but in the text is after fourth source (line 39) link to the 27th source (line 53), then are 4th, 5th 6th and 25th source. Similarly in next text.

Sources 8-11 and 17 are not in the text! One of them is self-citation!

There are still problems with the mismatch between the figure 2 and its description - 2 responses at age 5 in the figure, 3 responses in the text, three responses at ages 18 and 19 in the figure, four in the description. The total of responses in the úicture is 36 but in the text below is written that the sample is 27.

line 289 Figure 1, but it is Figure 2

line 325 29 responses in the Figure 4. What was the sample size?

Author Response

(The authors gave the same response as above.)

Round 3

Reviewer 3 Report

Thank you for improving the article. Please fix these small errors:

line 68 too much brackets

I think that the sample is very limited but the results could be interesting for readers.

Author Response

Thank you very much.The Brackets were removed as suggested by the reviewer.This  can be visible on track changer document attached.

Kind Regards

Dr Moses Basitere